# Rescue of Infectious Sindbis Virus by Yeast Spheroplast-Mammalian Cell Fusion

**DOI:** 10.3390/v13040603

**Published:** 2021-04-01

**Authors:** Lin Ding, David M. Brown, John I. Glass

**Affiliations:** 1J. Craig Venter Institute, 4120 Capricorn Lane, La Jolla, CA 92037, USA; lding@jcvi.org; 2J. Craig Venter Institute, 9605 Medical Center, Rockville, MD 20850, USA; dbrown40@umd.edu

**Keywords:** positive-sense single strand RNA virus, Sindbis virus, reverse genetics, *Saccharomyces cerevisiae*, galactose induction, transformation-associated recombination (TAR) cloning, frozen yeast spheroplasts, in vitro transfection, PEG-mediated fusion, alphavirus

## Abstract

Sindbis virus (SINV), a positive-sense single stranded RNA virus that causes mild symptoms in humans, is transmitted by mosquito bites. SINV reverse genetics have many implications, not only in understanding alphavirus transmission, replication cycle, and virus-host interactions, but also in biotechnology and biomedical applications. The rescue of SINV infectious particles is usually achieved by transfecting susceptible cells (BHK-21) with SINV-infectious mRNA genomes generated from cDNA constructed via in vitro translation (IVT). That procedure is time consuming, costly, and relies heavily on reagent quality. Here, we constructed a novel infectious SINV cDNA construct that expresses its genomic RNA in yeast cells controlled by galactose induction. Using spheroplasts made from this yeast, we established a robust polyethylene glycol-mediated yeast: BHK-21 fusion protocol to rescue infectious SINV particles. Our approach is timesaving and utilizes common lab reagents for SINV rescue. It could be a useful tool for the rescue of large single strand RNA viruses, such as SARS-CoV-2.

## 1. Introduction

Sindbis virus (SINV), an alphavirus in the *Togaviridae* family, was first isolated in 1952 from *Culex* mosquitoes in Cairo, Egypt [1]. It is transmitted to humans through a mosquito bite and usually causes mild symptoms [2] including fever, rash, and arthritis [3,4]. The SINV genome, a 11.7 kb positive-sense single-stranded RNA, is encapsulated by capsid protein and packaged into an envelope derived from the host cell plasma membrane containing viral glycoproteins [5]. A full-length cDNA clone of SINV was made in 1987. It was linearized and served as template for SP6 RNA polymerase to produce infectious SINV genomic RNA using in vitro transcription [6,7]. SINV particles were rescued by transfection of the resulting mRNA in the BHK-21 cells. SINV mRNA-based reverse genetics helped advance research on many positive stranded RNA viruses and is used as a core component of RNA virus-based vectors [8]. Indeed, various recombinant SINV vector systems have been implicated in research and biotechnology development [6,7,9,10,11,12,13,14]. Many SINV strains were developed over the years [15,16], among which a nonvirulent recombinant SINV strain, dsTE12Q, was made by changing the 55th amino acid residue of E2 envelope glycoprotein to glutamine [17].

*Saccharomyces cerevisiae*, a genetically tractable budding yeast, is a useful tool for virus research [18] ranging from studying the functions of viral proteins and manipulating viral genomes [19] to producing infectious particles [20]. Yeast shuttle vectors that are capable of replication and selection in *Escherichia coli* and *S. cerevisiae*, such as yeast centromeric plasmids (YCps), are valuable vehicles used to introduce heterologous gene expression [21]. Assembly of large RNA virus infectious cDNA constructs used to mainly rely on restriction enzymes [22]. The development of Gibson Assembly [23] and transformation-associated recombination (TAR) cloning [24] made it possible to seamlessly assemble large DNA molecules from multiple fragments into YCps. In addition, yeast spheroplasts have been used to deliver functional DNA or mRNA to Chinese hamster ovary (CHO) cells [25] and murine embryonic stem cells [26]. In 2017, our lab used Gibson Assembly and TAR cloning to assemble a 152 kb DNA virus genome of herpes simplex virus 1 (HSV-1) as a YCp [27], and then fusion of yeast spheroplasts and mammalian cells for a viral reverse genetics reaction to efficiently produce infectious HSV-1 particles [27,28].

For positive-sense single-stranded RNA viruses, there are two key steps for successful viral reverse genetics to generate infectious virus particles with engineered genomes: production of genomic RNA and its delivery to susceptible cells. The former was often achieved by in vitro transcription (IVT) of DNA copies of viral genomes using T7 RNA polymerase [29,30,31,32] or SP6 RNA polymerase [33,34,35]. The latter was usually achieved by transfecting the genomic RNA using electroporation or cationic detergent transfection reagents, such as Lipofectamine. One challenge with IVT is that both T7 and SP6 polymerases have high error rates (two errors per 10 kb [36] and one error per 10 kb to 30 kb [37], respectively). This potential problem with IVT can be avoided by transcribing viral genomic RNA in cellular systems where the RNA polymerase error rates are much lower. For instance, the yeast RNA polymerase II error rate is only four errors per million bases [38]. In many applications, reverse genetic cDNA constructs under promoters for specific cell lines (such as the cytomegalovirus (CMV) promoter for mammalian cells) are used instead of IVT [39]. However, templates encoding viral genomes still need to be introduced into susceptible cells via transfection, which can result in activation of antiviral mechanisms of the innate immune system that can defeat the whole process and require expensive transfection reagents that are subject to batch-to-batch variation.

In this study, we used the biotechnology workhorse organism, *S. cerevisiae*, to bypass problems of viral reverse genetics encountered when using IVT or transfection of viral genomes cloned as cDNA into mammalian cells. To produce infectious SINV mutants, we first generated a series of yeast centromeric plasmids (YCps) that are capable of expressing foreign genes. Then we inserted SINV-based viral genomes in a YCp under the control of a galactose-inducible promoter. The Sindbis viral genomes were transcribed by endogenous yeast RNA polymerase II. We delivered the SINV RNAs to BHK-21 cells using a yeast-mammalian cell fusion procedure. Taken together, we developed a powerful alternative system for production and functional studies of RNA viruses that would likely facilitate research and biotechnological applications using viruses with large RNA genomes, like coronaviruses (~30 kb).

## 2. Materials and Methods

### 2.1. Mammalian Cell Culture and Transfection

BHK-21 cells (ATCC^®^ CCL-10™, American Type Culture Collection, Manassas, VA, USA) were cultured in Dulbecco’s modified Eagle’s medium (DMEM) (Thermo Fisher Scientific 11995, Waltham, MA, USA), containing 10% heat inactivated Fetal Bovine Serum (FBS) (Thermo Fisher Scientific 16140071, Waltham, MA, USA) at 37 °C and 5% CO_2_. BHK-21 cells were transfected with pLDJIF15-SINV or dsTE12Q-GFP plasmid DNA using TransIT^®^-LT1 Transfection Reagent (Mirus Bio MIR2300, Madison, WI, USA). One microgram of plasmid DNA was used per transfection following manufacturer’s instruction (12-well plate).

### 2.2. Yeast Cells and Growth Media

Wild type yeast cells (W303α: *MATα ade2-1 ura3-1 his3-11 trp1-1 leu2-3 leu2-112 can1-00*) were grown in Difco™ YPD (BD 242810, Franklin Lakes, NJ, USA) for routine maintenance at 30 °C. Plasmid pLDJIF15 (or derivative) containing yeast cells were grown in CM Glucose Broth w/o Tryptophan (-TRP) (Teknova C71731, Hollister, CA, USA) at 30 °C. Plasmid pLDJIF19 (or derivative) containing yeast cells were grown in CM Glucose Broth, Dry, Adenine-60, without histidine (-HIS) (Teknova C7112, Hollister, CA, USA) at 30 °C. Plasmid pLDJIF20 (or derivative) containing yeast cells were grown in CM Glucose Broth, Dry, w/o uracil (-URA) (Teknova C7141, Hollister, CA, USA) at 30 °C. All yeast liquid cultures were grown overnight with 220 revolutions per minute (rpm) agitation.

### 2.3. Plasmids

Gibson Assembly (New England Biolabs E2611, Ipswich, MA, USA) was used to generate all plasmids (Table 1). Fragments, if not described in text, were amplified with 60 bp primers that overlap neighboring fragments by 30 bp. Sequences (GenBank files) of plasmids can be found in the supplementary files. Primers used for this study are listed in Table 2.

### 2.4. Yeast Transformation

Yeast centromeric plasmids (YCps) were transformed via electroporation using 0.2 cm gap Gene Pulser/MicroPulser Electroporation Cuvettes (Bio-Rad 1652086, Hercules, CA, USA) and a Gene Pulser Xcell™ Electroporation System using a protocol described by Becker and Lundblad [40]. Visible transformants appeared in 2 to 3 days on selective agar plates. See Table 3 for yeast strains used in this study.

### 2.5. Yeast Galactose Induction

Five mL overnight yeast liquid cultures in selective medium were used for galactose induction. Prior to the induction, yeast cultures were harvested by centrifugation at 1600× *g* for 5 min and washed once with an equal volume of sterile water. Galactose induction was done by resuspending the washed yeast cell pellet in 50 mL of YPG medium (10g/L Bacto™ yeast extract (Thermo Fisher Scientific 212750, Waltham, MA, USA), 20 g/L Bacto™ peptone (BD 211677, Franklin Lakes, NJ, USA), 100 mL 20% galactose (*w*/*v*) (MilliporeSigma G5388, Burlington, MA, USA)) grown at 30 °C with 220 rpm agitation for 5 h.

### 2.6. Sindbis Virus

All experiments were conducted in a biosafety level 2 (BSL2) certified lab space. Green fluorescent protein (GFP) tagged Sindbis virus (dsTE12Q-GFP) infectious cDNA construct was a gift from Dr. Reed Shabman. The GFP coding sequence was inserted at the BstEII site in the original dsTE12Q [34] cDNA construct.

### 2.7. Yeast Spheroplasts Preparation and Storage

Galactose-induced yeast liquid cultures (50 mL in a 250-mL flask) were collected in 50 mL conical tubes and harvested by centrifugation at 1600× *g* for 5 min at 4 °C then washed once with 50 mL sterile water. The pellet was resuspended in 20 mL of 1 M filter-sterilized sorbitol and incubated on ice for at least 4 h (up to 24 h). The mixture was inverted occasionally during the incubation to prevent cell settling. After incubation, cells were pelleted at 1600× *g* for 5 min at 4 °C and resuspended in 20 mL of SPEM solution (1 M sorbitol, 10 mM EDTA pH 8, 2.08 g/L Na_2_HPO_4_•7H_2_O and 0.32 g/L NaH_2_PO_4_•1H_2_O). Forty µL of 14 M β-mercaptoethanol was added to the suspension and followed by mixing by inversion. Forty µL of 10 mg/mL Zymolyase-20T (Amsbio 120491-1, Milton Park, Abingdon, UK) solution (200 mg Zymolyase-20T, 9 mL of sterile water, 1 mL of 1 M Tris-Cl pH 7.5 and 10 mL of 50% glycerol; aliquoted and stored at −20 °C) and mixed by inversion. The mixture was incubated at 30 °C for 25 min with 50 rpm agitation. After digestion, 100 µL of the mixture was mixed with either 900 µL of 1 M sorbitol or 2% SDS solution and the OD_600_ was measured. To ensure good spheroplast preparation, digestion continued until the ratio of the measurements mentioned above was about three. We added 30 mL of 1 M sorbitol to the cell suspension and mixed by gentle inversion. The spheroplasts were pelleted by centrifugation at 1600× *g* for 5 min at 4 °C. The pellet was resuspended in 20 mL 1 M sorbitol gently with a 25-mL pipet, then an additional 30 mL 1 M sorbitol was added. The cell suspension was mixed by inversion. The spheroplasts were centrifuged at 1600× *g* for 5 min at 4 °C. We resuspended the pellets in a storage solution (1 M sorbitol + 15% DMSO, 10^8^ cells/mL, 300 µL/1.5-mL tube) and stored them at −80 °C for up to a year [41].

### 2.8. Yeast to BHK-21 Cell Fusion

BHK-21 cells (70–80% confluence) were harvested from T75 flasks, washed twice with 10 mL DMEM + 10% FBS, then diluted to 2 × 10^5^ cells/ ml. Frozen yeast spheroplasts were thawed on ice for about 5 min prior to fusion. We then added 33 µL of 10× TC (100 mM Tris-HCl pH 7.5, and 100 mM CaCl_2_) and mixed spheroplast suspension by gentle inversion. The spheroplasts were centrifuged at 1600× *g* for 5 min at room temperature, washed with 500 µL of a STC (1 M sorbitol, 10 mM Tris-HCl pH 7.5, and 10 mM CaCl_2_) solution, and resuspended in 300 µl STC (10^7^ spheroplasts/30 µL). We then added 500 µL of BHK-21 cells (10^5^ cells/fusion) to 1.5-mL tubes, followed by addition of 30 µL of the spheroplast-STC suspension. BHK-21 cells and yeast spheroplasts (1:100) were mixed by inversion, incubated for 5 min at room temperature, then pelleted at 2300× *g* for 30 s. The pellet was resuspended in 450 µL of freshly prepared 44% PEG 1500 (MilliporeSigma 81210, Burlington, MA, USA) solution (dissolved in 75 mM HEPES and 10 mM CaCl_2_) and 50 µL DMSO (MilliporeSigma D2650, Burlington, MA, USA), and then incubated for 45 s at room temperature. One milliliter of DMEM was added in the tube at the end of the incubation. Cells were pelleted at 2300× *g* for 30 s and washed with 1 mL DMEM before being resuspended in 1 mL DMEM + 10% FBS + 100 U/mL of Penicillin/Streptomycin/Amphotericin B Solution (MilliporeSigma 516104-20ML-M, Burlington, MA, USA). This mixture was added to a well in a 6-well plate which already contained 2 mL of the medium for final resuspension. The medium was replaced after BHK-21 cells attached to the bottom of the well (usually about 2 h).

### 2.9. Sindbis Virus Growth Curve and Titer Evaluation via Plaque Assay

Supernatant from transfection or the fusion experiment was collected at the indicated timepoints and stored at −80 °C. We determined viral titers by plaque assays done in triplicate in 6-well plates. BHK-21 cells (7.5 × 10^5^) were seeded in 1.5 mL of DMEM + 10% FBS (per well). The plates were incubated at 37 °C and 5% CO_2_ for 1 day. Collected viral samples were thawed in a 37 °C water bath and diluted in DMEM + 2% FBS (1:10 serial dilutions). Prior to inoculation, growth medium was removed. BHK-21 cells were washed once with 2 mL PBS and inoculated with diluted samples (400 µL/well). The plates were incubated at 37 °C and 5% CO_2_ for 1 h with gentle agitation every 15 min. After the adsorption period, 2 mL of pre-warmed (56 °C) immobilizing medium (1:1 mixture of DMEM + 2% FBS and 0.6% agarose solution) was added to each well. After 3 days of incubation at 37 °C and 5% CO_2_, the overlay was removed; the wells were washed with 2 mL PBS and stained with 2 mL of a crystal violet solution (2 g crystal violet, 60 mL absolute ethanol, 40 mL formaldehyde, and 100 mL PBS) at room temperature in the dark for 30 min. After incubation, crystal violet solution was aspirated, and the plates were washed with tap water and air-dried. Visible plaques were counted and used to estimate viral titers.

### 2.10. Microscopy

We used a Celigo Imaging Cytometer (Nexcelom Bioscience, Lawrence, MA, USA) and AxioCam ERc 5s Microscope (Zeiss, White Plains, NY, USA) to acquire fluorescence images. The images were processed with manufacturer provided software.

## 3. Results

### 3.1. Yeast Centromeric Plasmids (YCps) for Galactose Induction in Saccharomyces Cerevisiae

We designed a YCp (pLDJIF15, Figure 1A) for heterologous gene expression under control of yeast transcriptional machinery. It featured a galactose inducible promoter, P_Gal1_, that allowed controlled mRNA expression by yeast RNA polymerase II for any DNA sequence inserted between the two 30 bp adapters (Figure 1B), namely Adapter 1: 5′-AGAACCCACTGCTTACTGGCTTATCGAAAT-3′ and Adapter 2: 5′-CTGTGCCTTCTAGTTGCCAGCCATCTGTTG-3′. Any DNA fragment flanked by Adapters 1 and 2 can be inserted into the backbone that is PCR amplified from pLDJIF15 with primers D199 and D200 (Figure 1B, Table 1). We used a green fluorescent protein (GFP) gene that was codon-optimized for yeast expression (yeGFP) to test the requirement of galactose induction in plasmid. The open reading frame (ORF) of yeGFP was amplified with two 60 base primers (D205 and D201, Appendix AA, Table 1) that consisted of 30 bases of the adapter sequence and 30 bases of the 5′ or 3′ end of the yeGFP ORF. The insert was assembled into pLDJIF15 using Gibson Assembly (Figure 2A). The resulting plasmid, pLDJIF15-yeGFP, was confirmed by colony PCR with primers, D66 and D67, that flanked the adapters (Appendix AA, Table 1). We transformed the plasmid into yeast cells and induced the yeGFP expression with galactose-containing growth medium (Figure 2A, steps 1–4). After 5 h, a robust GFP signal was observed in yeast cells grown in the galactose-containing medium, while no GFP signal was observed in uninduced yeast cells (Appendix AB). Our result suggested that the adaptors did not interfere with the galactose induction of yeGFP expression. Thus, we concluded pLDJIF15 was suitable for controlled expression of foreign genes.

pLDJIF15 also consisted of two copies of yeast codon-optimized red fluorescent protein (yeCherry) under the GAP/GDP promoter [42] (Figure 1A). The two copies resulted in high levels of constitutive yeCherry expression. Yeast cells transformed with pLDJIF15 or derivatives had an enhanced red fluorescent (RFP) signal regardless of whether they were induced or uninduced (Appendix AB). Moreover, successful transformants were pink (Figure 2 yeast pellet), allowing for easy selection.

Between the two adapters, we introduced a yeast translational entry site or Kozak sequence (reviewed in [43]) and a sequence encoding a Myc tag. This allows pLDJIF15 to be used as a tagging vector, where a transgene can be inserted in frame (instead of using the two adapters) behind the Kozak sequence and in front of the Myc tag. Two additional plasmids, pLDJI19 and pLDJIF20, were generated by replacing the *TRP1* gene (auxotrophic marker) in pLDJIF15 with *HIS3* and *URA3*, respectively (Figure 1A). These three YCps make protein co-expression possible.

### 3.2. Yeast Cytoplasmic Content Can Be Transferred to BHK-21 Cells via PEG Mediated Fusion

To develop a PEG-mediated yeast-BHK-21 cell fusion procedure, we used yeast cells that constitutively express yeCherry from pLDJIF15 (Appendix AB). The cytoplasmic localized yeCherry serves as an indicator for yeast cytoplasmic contents. Therefore, by monitoring the RFP signal, we are able to determine the efficiency of the mixing of the cytoplasms.

We treated the yeast cells (yLDJIF9) harboring pLDJIF15 with Zymolyase to permeabilize the yeast cell walls and made frozen cells spheroplasts. We adapted a yeast-mammalian cell fusion protocol established by David Markie for yeast-CHO fusion [25]. After optimization, our result suggested that incubating yeast spheroplasts and BHK-21 cells in a 100:1 ratio with a mixture of 44% PEG and 10% DMSO at room temperature for 45 s produced about 5% fused cells (Figure 2B). This is consistent with what we observed for HSV-1 DNA genome delivery in unsynchronized mammalian cells [28] and was sufficient for our purpose.

### 3.3. Infectious Sindbis Virus Is Produced via Yeast to BHK-21 Cell Fusion

Our Sindbis virus cDNA construct [34] contained an SP6 promoter upstream of the viral genome. The construct was originally designed to serve as a template to produce infectious Sindbis genomic RNA by in vitro transcription after the plasmid was linearized at an XhoI site [34,35]. We obtained a modified version of this construct in which a copy of GFP ORF (codon optimized for mammalian cell expression) was inserted at a BstEII site. We found a single restriction site (SacI) upstream of the SP6 promoter. Together with XhoI, we were able to release the entire GFP tagged Sindbis genome with SP6 promoter and inserted it between the adapters in pLDJIF15 (Figure 3A) backbone that was amplified with primers (D250 and D251; Figure 3A, Table 1) in a single piece. Since no PCR amplification was involved in producing the SINV coding sequence, SINV genome integrity in the resulting plasmid, pLDJIF15-SINV, was unlikely to be comprised. To our surprise, both pLDJIF15-SINV and the parental plasmid, dsTE12Q-GFP, were able to rescue the Sindbis virus when directly transfecting the plasmid DNA to BHK-21 cells (Appendix A), despite lacking a CMV promoter for mammalian mRNA expression [44].

Our initial plan was to co-transform pLDJIF15-SINV with a galactose-inducible SP6 RNA polymerase expression plasmid using pLDJIF19 or 20 as a backbone into the same yeast cell. The rational was that by controlling the expression of SP6 RNA polymerase, we could control the Sindbis genomic RNA expression in yeast cells. Price et al. suggest that it is possible to induce flock house virus (FHV) genomic mRNA expression under Gal1 promoter in yeast without SP6 RNA polymerase [19]. Indeed, we found that pLDJIF15-SINV by itself responded to galactose-induced Sindbis genomic mRNA expression (Figure 3). We prepared frozen spheroplasts from yeast cells harboring pLDJIF15-SINV (yLDJIF21) grown in galactose (induced) or glucose (uninduced) containing medium. The frozen spheroplasts were thawed carefully and fused with BHK-21 cells. After fusion, we observed the GFP signal (on Day 2), followed by the cytopathic effect (CPE) (on Day 3) from BHK-21 cells fused with spheroplasts grown in galactose (induced; Figure 3B images 1, 3, 5) but not glucose (uninduced; Figure 3B images 2, 4, 6) medium (Figure 3B). We measured SINV viral titer from the supernatant (Figure 3C). After fusing the spheroplasts made from yeast (yLDJIF21) grown in galactose medium, we detected viral production on day 1 after fusion (4.2 × 10^3^ pfu/mL) via plaque assay (Figure 3B,C). The titer increased to 3.3 × 10^8^ pfu/mL on day 2 after fusion and peaked (7.3 × 10^8^ pfu/mL) on day 3 after fusion when accompanied with CPE (Figure 3B,C). The peak SINV titer produced from our procedure was comparable to the titer generated from transfecting BHK-21 cells with in vitro transcribed SINV genome [35]. The successful rescue of SINV was also confirmed by the increase in the GFP signal (Figure 3B). Importantly, no viral titer was observed in BHK-21 cells fused with spheroplasts prepared from uninduced yLDJIF21. This indicated it was the mRNA expression driven by GAL1 promoter, rather than the pLDJIF15-SINV DNA, that rescued SINV.

## 4. Discussion

SINV, one of the most widely distributed alphaviruses [4], is a well-studied virus with a positive-sense single-stranded RNA genome [5,45]. SINV-based vector systems are routinely used as expression vectors for heterologous gene expression and infectious particles production [39,46]. They have potential applications in gene therapy [13,47,48] and vaccine development [7]. However, SINV (strain-specific or recombinant) production from cloned genomes largely relies on IVT followed by transfection of the resulting RNA genome [6,7,10]. Reverse genetics for viruses with RNA genomes, including SINV, using IVT in general are susceptible to unintended outcomes caused by the high error rates of commonly used bacteriophage RNA polymerases [36,37], batch-to-batch variation, and short shelf-life of transfection reagents, such as Lipofectamine.

We developed an alternative way to produce SINV particles that used high fidelity cellular RNA polymerases to generate viral genomic RNA and by using direct yeast-to-mammalian cell transfer of that RNA. The direct cell-to-cell RNA transfer avoids potential problematic issues of RNases and ineffective transfection reagents that plague transfection of naked RNA. For this new approach, we made a series of improved galactose-inducible YCps, pLDJIF15, 19, and 20, with different auxotrophic markers capable of containing templates for viral RNA transcription (Figure 1). Multiple markers enabled us to express more than one gene in a single yeast. Our YCps serve as a backbone for inducible expression of tagging proteins or libraries through Gibson Assembly (Figure 2Ai) or TAR cloning (Figure 2Aii). The latter was crucially important for assembly of large sequences. The yeCherry was constitutively expressed in yeast cells under P_GDP_. It is a good indicator for successful yeast cell transformation and yeast-mammalian cell fusion.

We generated an SINV cDNA construct, pLDJIF15-SINV, from a well-established strain (dsTE12Q-GFP) by Gibson Assembly. Initially, our plan was to control SINV genomic RNA expression from pLDJIF15-SINV by inducing SP6 RNA polymerase expression from a co-transformed plasmid. Surprisingly, we found that although lacking a mammalian expression promoter (such as a CMV promoter), both dsTE12Q-GFP, routinely used as template for IVT [35], and its derivative pLDJIF15-SINV, when transfected into BHK-21 cells, generated SINV virus particles (Appendix A). This may be worth further investigation.

We show, for the first time, that yeast spheroplast-mammalian cell fusion rescues SINV particles. Our method only requires installation of the yeast cell contents into the mammalian cell cytoplasmic to produce the virus. This strategy is applicable to the rescue other single-stranded RNA viruses, especially for the ones with large genomes like coronaviruses. The genome size of SARS-CoV-2, the coronavirus that caused the COIVD-19 pandemic, is approximately 30 kb, and known to be the largest RNA virus genome. Its infectious cDNA clone was constructed from pieces and assembled into a single bacterial artificial chromosome (BAC) under a CMV promoter with restriction enzymes digestions and ligations [49]. The resulting BAC was transfected into Vero E6 for virus rescue. Recently, Thao et al. assembled the SARS-CoV-2 genome in a YCp under a T7 promoter using TAR cloning [50]. Our approach (Figure 2Aii) suggests an alternative way to deliver variants SARS-CoV-2 RNA genome directly into Vero E6 cells to facilitate virus production. For other RNA viral genomes, such as most Baltimore classification’s Group V viruses that require an additional helper protein/s to become infectious [51], the helper protein or additional RNA molecules that encode that protein could also be expressed in yeast and delivered via yeast fusion from a single or multiple yeast. Hence our system has broad implications to a diverse array of viruses.

Our method demonstrates several throughput improvements over conventional viral reverse genetics methods. We use yeast as a viral genome delivery vehicle. Yeast is a widely used model organism that has facilitated investigations of several viruses that attack higher eukaryotes [18]. Moreover, yeast spheroplast has become a popular vehicle to deliver its cytoplasmic content that contains heterologous expressed material (protein, DNA, and RNA) into mammalian cells [25,26,28,52]. However, preparing yeast spheroplasts every time to delivery virus genomic RNA could be cumbersome. We adopted a simple procedure to store frozen yeast spheroplasts. It dramatically shortened the time for fusion experiments and made this system suitable for high throughput screening purposes. Furthermore, our yeast fusion-based delivery system is successfully linked with the bottom-up synthetic biology tools of yeast [53]. Our improvements enable researchers to build synthetic copies of large RNA viral genome in yeast, then deliver it to the desired mammalian cell. In the future, this could be used in conjunction with protein and DNA delivery.

Additionally, our system can serve as a reporter system to study inhibitors of mammalian RNA interference (RNAi). BHK-21 cells lack interferons (a major innate immunity pathway in mammalian cells), therefore were used to uncover the RNAi mechanism [54]. Using our system, one could screen for mammalian RNAi inhibitors by expressing proteins of interest in yeast (the same yeast cell that harbors pLDJIF15-SINV or a separate one) and then fusing the yeast spheroplasts (one population or a mixture) with BHK-21 cells. An elevated SINV viral titer is indicative of anti-RNAi activity of that protein. The fusion rate based on the yeCherry protein transferred from yeast to BHK-21 cell was approximately 5%. Increased sensitivity of this reporter system can be achieved by enriching the population of RFP positive BHK-21 cells right after fusion with Fluorescence-activated cell sorting (FACS). Measuring SINV titer via plaque assay is not ideal for high throughput screening. The SINV strain that we used produced CPE. When that occurred, the intensity of the GFP signal was no longer corelated with the SINV viral titer. Agapov et al. suggested that a single amino acid change of the SINV NSP2 protein (P726L) renders SINV non-cytopathic [55]. It is foreseeable that by switching to that SINV strain instead of dsTE12Q, our system could be a powerful RNAi screening platform.

Herein we present what we believe to be a superior approach to doing viral reverse genetics work with SINV. Critically, anyone using our approach to produce SINV would need to do all experimental steps that involve potential production of SINV in a BSL-2 or higher laboratory. Furthermore, while SINV is not a significant human pathogen and our work with this virus poses a few concerns about possible misuse of this new technology, we must acknowledge that advancement of reverse genetics for viruses with RNA genomes does pose some dual use risks. The key advantage of our approach over more conventional RNA virus reverse genetics approaches that generate viral RNA genomes using IVT is that our approach is more likely to produce a virus with the designed genome sequence. In our view, this advantage outweighs any dual use risks that might arise by publishing this technique.

## Figures and Tables

**Figure 1 viruses-13-00603-f001:**
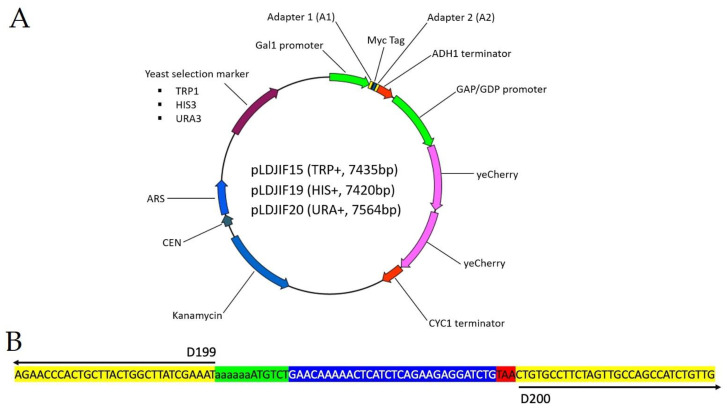
New yeast centromeric plasmids (YCps) for galactose inducible expression. (**A**) A schematic of three yeast centromeric plasmids. Not drawn to scale. See GenBank files in supplementary data for sequences. (**B**) Detailed sequence detail from Adapter 1 and 2. Yellow, left: Adapter 1, right: Adapter 2; Green: Yeast Translational Entry Site; Blue: Myc Tag; Red: Stop codon.

**Figure 2 viruses-13-00603-f002:**
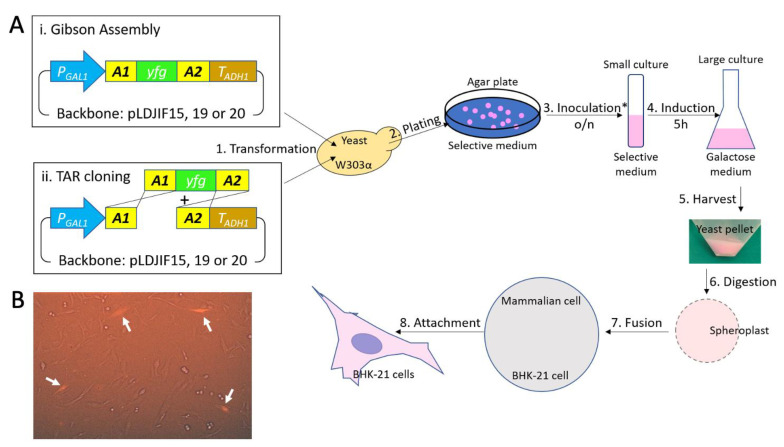
Yeast cytoplasmic elements are transferred to mammalian cells (BHK-21) via fusion. (**A**) An outline of yeast to mammalian cell fusion procedure (from sequence to fusion). * Colony PCR screening was used if transformation-associated recombination (TAR) cloning (ii) pathway was used. (**B**) Fluorescence micrographs of BHK-21 cells fused with yeast spheroplast prepared from induced yLDJIF9 (W303α transformed with pLDJIF15). yeCherry (pink) was expressed in yeast only. Pink BHK-21 cells (pointed out with an arrow) appear when the cytoplasmic element, including yeCherry, is transferred from yeast to BHK-21 cells via fusion.

**Figure 3 viruses-13-00603-f003:**
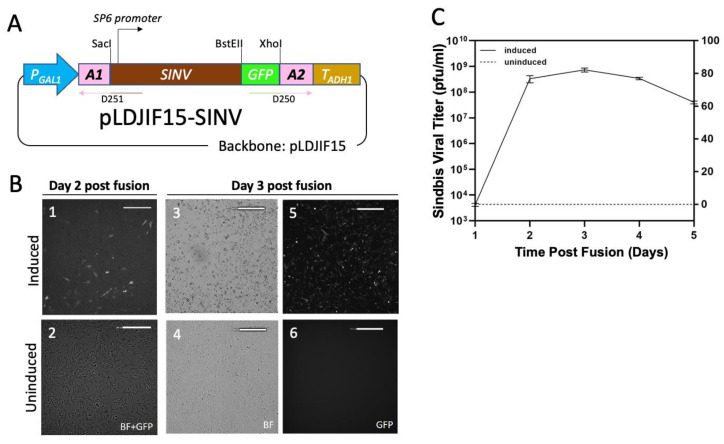
Sindbis virus was rescued by yeast to mammalian cell fusion. (**A**) A schematic of pLDJIF15-SINV, a green fluorescent protein (GFP)-tagged infectious cDNA clone of Sindbis virus (SINV). See GenBank files in supplementary data for sequences. (**B**) Fluorescence micrographs of fusion experiments taken on day 2 and 3 post fusion experiments. yLDJIF21 (W303α transformed with pLDJIF15-SINV) was grown overnight in −TRP medium was used to inoculate −TRP medium (uninduced) or YPG medium (induced) for 5 h at 30 °C. Cytopathic Effect (CPE) was observed on day 3 post fusion experiment with galactose-induced yeast cells. Representative fields; scale bar: 500 µm. (**C**) Growth curve of SINV rescued from yeast-mammalian fusions. SINV viral titers from day 1 to 5 after fusion were measured by plaque assay. Rescue of SINV viral particle was observed from spheroplast prepared from galactose-induced yeast culture. (Day 1: 4.2 × 10^3^ pfu/mL; Day 2: 3.3 × 10^8^ pfu/mL; Day 3: 7.3 × 10^8^ pfu/mL; Day 4: 3.4 × 10^8^ pfu/mL; Day 5: 4 × 10^7^ pfu/mL). No SINV was detected from uninduced yeast culture. Induced: Left Y-axis (log scale). Uninduced: Right Y-axis (linear scale).

**Table 1 viruses-13-00603-t001:** Plasmids.

Plasmid Name	Purpose	Source
dsTE12Q-GFP	GFP tagged Sindbis genome (*dsTE12Q*) an infectious cDNA clone for in vitro transcription	A gift from Dr. Reed Shabman
pLDJIF15	Galactose inducible mRNA or protein, *TRP1* (GenBank accession MW820849)	This study
pLDJIF15-yeGFP	Galactose inducible yeGFP, *TRP1* (GenBank accession MW82085)	This study
pLDJIF19	Galactose inducible mRNA or protein, *HIS3* (GenBank accession MW820851)	This study
pLDJIF20	Galactose inducible mRNA or protein, *URA3* (GenBank accession MW820852)	This study
pLDJIF15-SINV	Galactose inducible dsTE12Q-GFP mRNA, *TRP1* (GenBank accession MW820850)	This study

See GenBank files supplementary for full sequences.

**Table 2 viruses-13-00603-t002:** Primers.

Primer Name	Sequence 5′→3′	Purpose
D66	CAACCATAGGATGATAATGCGATTAG	Testing for insertion between Adapter 1 and 2
D67	TGAGAAAGCAACCTGACCTACAG	
D199	**ATTTCGATAAGCCAGTAAGCAGTGGGTTCT**	Amplify pLDJIF15, 19, and 20 with adapter 1 and 2
D200	**CTGTGCCTTCTAGTTGCCAGCCATCTGTTG**	
D205	**AGAACCCACTGCTTACTGGCTTATCGAAAT**ATGGTTAGTAAAGGTGAAGAATTATTCACT	Amplify yeGFP with adapter 1 and 2
D201	**CAACAGATGGCTGGCAACTAGAAGGCACAG**TTATTTGTACAATTCATCCATACCATGGGT	
D250	aaaaaaaaaaaaaaaggggaattcctcgag**CTGTGCCTTCTAGTTGCCAGCCATCTGTTG**	Amplify pLDJIF15 backbone to insert SacI-SINV-GFP-XhoI cassette through Gibson Assembly
D251	gttctaacgacaatatgtccatacgagctc**ATTTCGATAAGCCAGTAAGCAGTGGGTTCT**	

Bold: adapter sequences; underline: restriction sites (XhoI: ctcgag; SacI: gagctc) Lower case: 5′ or 3′ end of Sindbis virus fragment released from XhoI and SacI digestion of plasmid dsTE12Q-GFP.

**Table 3 viruses-13-00603-t003:** Yeast Strains.

Name	Genotype	Source
W303α	*MATα ade2-1 ura3-1 his3-11 trp1-1 leu2-3 leu2-112 can1-00*	ATCC^®^ 208353™
yLDJIF9	*MATα ade2-1 ura3-1 his3-11 trp1-1 leu2-3 leu2-112 can1-00 [pLDJIF15]*	This study
yLDJIF21	*MATα ade2-1 ura3-1 his3-11 trp1-1 leu2-3 leu2-112 can1-00 [pLDJIF15-SINV]*	This study
yLDJIF22	*MATα ade2-1 ura3-1 his3-11 trp1-1 leu2-3 leu2-112 can1-00 [pLDJIF15-yeGFP]*	This study

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
