# Peer review of "Rescue of Infectious Sindbis Virus by Yeast Spheroplast-Mammalian Cell Fusion"

_viruses, 2021, doi:10.3390/v13040603_

Round 1

Reviewer 1 Report

This novel study by Ding et al uses yeast spheroplasts to produce infectious Sindbis virus, using yeast centromeric plasmids.  By transcribing this viral genome with RNA polymerase II it leads to a much lower error rate than with IVT using T7 or SP6 polymerases. However, it doesn’t overcome the inefficiency of electroporating the YCps into yeast cells, and this should be clarified. The yeast cytoplasm is then transferred to BHK cells with PEG, and this has an efficiency of only 5%.  Is this a problem for SINV yield?   If this efficiency was increased much higher viral titres could be obtained.  Please address this in your results in lines 301-305. Interestingly, the virus mRNA was transcribed by the Gal 1 promoter without the need to add SP6 polymerase.  This novel technique has important biotechnology implications with applications for virus and vaccine production.   The method is well described and would be easy to reproduce from the description. Therefore I think that with the style changes made below, this work is suitable for publication

Comments

There are many English grammar, style and spelling mistakes throughout.

Abstract. 

Line  11, remove the.  .

Line 14… remove followed by transfection

Line 15 remove the

Line 18 fusion with BHK cells.

Line 19.  Remove only

Introduction.

Line 34 Transfection into what cells?

Line 38 Developed, not discovered

Line 76 in one of YCps- grammar

Results

Line 208 and 209 Adaptor sequence should be moved to Methods

Line 209-211 incorrect grammar

Line 213        remove out

Line 217        colony PCR confirmed is incorrect style/grammar

Line 219        performed galactose induction- incorrect grammar

Line 237        myc tagging vector- change style

Figure 2       Yeast cytoplasmic elements can be transferred to mammalian cells (BHK-21) via fusion- incorrect grammar- replace “can be” with “are”: they either are or they aren’t, it’s not conditional.

Colony PCR screening is needed if TAR cloning (ii) 244 pathway is used- grammar and tense

Discussion

Needs close reading for typographical and grammatical errors

Author Response

We thank reviewer 1 for their comments. Reviewer 1 said we need to overcome the inefficiency of electroporating the YCps into yeast cells. While our process does not involve electroporation, we acknowledge that only about 5% of the treated BHK cells end up producing Sindbis virus. For simple production of virus for use in other processes, the 5% efficiency is completely acceptable. For other purposes, such as studies of the effects of the insertion of viral genomic RNA into cells, the 5% yeast:mammalian cell rate is not ideal, but it is comparable  to other research publications. We addressed this both in the Results section as requested by Reviewer 1 and in the Discussion. In Discussion we noted that we think by using FACS (text suggesting this added to the revised manuscript on lines 383-384), we could enrich the fused cell population and increase the sensitivity. 

Reviewer 1 also pointed out a series of grammar, spelling, and style mistakes. These have all been addressed as listed here:

Abstract

Line  11, remove the.  .

-We removed the “the”.

Line 14… remove followed by transfection

-We removed the phrase.

Line 15 remove the

-We removed the “the”.

Line 18 fusion with BHK cells.

-The relevant sentence was changed to "Using spheroplasts made from this yeast, we established a robust polyethylene glycol-mediated yeast:BHK-21 fusion protocol to rescue infectious SINV infectious particles

Line 19.  Remove only

-We removed the “only”.

Introduction.

Line 34 Transfection into what cells?

-We added BHK-21 cells.

Line 38 Developed, not discovered

-We changed discovered to developed.

Line 76 in one of YCps- grammar

-We fixed the phrase.

Results

Line 208 and 209 Adaptor sequence should be moved to Methods

-We thank the reviewer for the suggestion. Yet we think the sequences are important as it is a part of how we developed the plasmid shown in Figure 1B. The primers used in the procedure are listed in Methods.

Line 209-211 incorrect grammar

-We fixed the sentence.

Line 213        remove out

-We removed the word.

Line 217        colony PCR confirmed is incorrect style/grammar

-We fixed the grammar.

Line 219        performed galactose induction- incorrect grammar

-We fixed the grammar/style.

Line 237        myc tagging vector- change style

-We fixed the style.

Figure 2       Yeast cytoplasmic elements can be transferred to mammalian cells (BHK-21) via fusion- incorrect grammar- replace “can be” with “are”: they either are or they aren’t, it’s not conditional.

-We thank the reviewer for pointing it out and we fixed the grammar.

Colony PCR screening is needed if TAR cloning (ii) 244 pathway is used- grammar and tense

-We fixed the grammar and tense.

Discussion

Needs close reading for typographical and grammatical errors

-We did proof-reading and corrected errors.

Reviewer 2 Report

Ding et al reported a novel reverse genetic method to recover infectious SINV by yeast spheroplast-mammalian cell fusion. There are several advantages of the technique: 1) the expression of viral genomic mRNA with higher fidelity with using Yeast RNA polymerase instead of using in vitro translation; 2) the potential for packaging large genome in yeast transfer vector for virus rescue and 3) no need for the conventional plasmid preparation in bacteria  to which some genes are toxic and not stable. In short, this new method has the potential to improve the efficiency for the rescue of some large or difficulty RNA viruses.

Though the method is efficient in recover SINV, it is noted that only 5% of mammalian cells were fused and thus the transfection rate is far too low for the application of RNAi research. I would think that authors overestimate the potential use in the field.

SINV is a positive single-strand RNA virus, and its mRNA-like genome is infectious. It is no surprise to understand that the transcript of viral genome by yeast polymerase would be also infectious even though SP6 RNA polymerase was not involved. Author should rewrite this part in discussion section.

Other minor issues:

  1. Line 106. The sequence file of the plasmid is missing in the supplementary files.
  2. Line 207, figures 1, 2 and 3. The abbreviation for adaptor 1 and 2 (A1 and A2) should be also marked in Figure 1.
  3. Line 274 and figure 3A. The plasmid pLDJIF16 should be renamed to reflect the insertion of SINV genome so as to be distinguishable from the empty vector series.
  4. Line 292. Rephrase the sentence “Price et al., suggest that ... induce flock house virus (FHV) expression under Gal1 promoter in yeast without SP6 RNA polymerase”. A virus is rescued from the expressed viral genomic like-mRNA.
  5. Page 8. The image panels in figure 3B are in poor quality. Images with higher resolution and the enlarged or cropped areas would improve the demonstration of fluorescence and CPE after yeast-BHK 21 cell fusion.

Author Response

We thank Reviewer 2 for their careful reading of our manuscript and logical suggestions. We hope we have answered the concerns and fixed the writing.

Concerning this Reviewer 2 comment: Though the method is efficient in recovering SINV, it is noted that only 5% of mammalian cells were fused and thus the transfection rate is far too low for the application of RNAi research. I would think that authors overestimate the potential use in the field.

-We are using the system as a way of screening molecules that are inhibitors of mammalian RNAi pathway.

The fusion rate is low but it is comparable to current publications. In addition to refining the fusion conditions, we could enrich the fused cell population (RFP+) by FACS. A comment to this effect is added in the Discussion (lines 383-384).

Reviewer 2 wrote: SINV is a positive single-strand RNA virus, and its mRNA-like genome is infectious. It is no surprise to understand that the transcript of viral genome by yeast polymerase would be also infectious even though SP6 RNA polymerase was not involved. Author should rewrite this part in discussion section.

-We agree that the there is no surprise that SINV genome transcribed by yeast polymerase is infectious. The novelty, as the reviewer pointed out is the streamline of the process. We addressed this in the discussion.

Other minor issues:

  1. Line 106. The sequence file of the plasmid is missing in the supplementary files.

-We included the GBK files. We will also submitted them to NCBI and plan to include accession numbers in our revised manuscript. We had expected GenBank to provide us accession numbers by now, but the process is taking much longer than advertised. We have contacted NCBI and hopefully can add the accession numbers for the 4 plasmids discussed in the manuscript when we fix galley proofs.

  1. Line 207, figures 1, 2 and 3. The abbreviation for adaptor 1 and 2 (A1 and A2) should be also marked in Figure 1.

-We marked A1 and A2 on the map.

  1. Line 274 and figure 3A. The plasmid pLDJIF16 should be renamed to reflect the insertion of SINV genome so as to be distinguishable from the empty vector series.

-We have made the corrections in text and figures.

  1. Line 292. Rephrase the sentence “Price et al., suggest that ... induce flock house virus (FHV) expression under Gal1 promoter in yeast without SP6 RNA polymerase”. A virus is rescued from the expressed viral genomic like-mRNA.

-We corrected sentence.  

  1. Page 8. The image panels in figure 3B are in poor quality. Images with higher resolution and the enlarged or cropped areas would improve the demonstration of fluorescence and CPE after yeast-BHK 21 cell fusion.

 -We improved the quality of the figure.